# Modality-Aware SAM:
# Sharpness-Aware-Minimization Driven Gradient Modulation for Harmonized Multimodal Learning

**Hossein Rajoli**
Holcombe Department of ECE
Clemson University
hrajoli@clemson.edu

**Jie Ji**
Holcombe Department of ECE
Clemson University
jji@clemson.edu

**Xiaolong Ma**
Department of ECE
University of Arizona
xiaolongma@arizona.edu

**Fatemeh Afghah**
Holcombe Department of ECE
Clemson University
fafghah@clemson.edu

## Abstract

In multimodal learning, dominant modalities often overshadow others, limiting generalization. We propose Modality-Aware Sharpness-Aware Minimization (M-SAM), a model-agnostic framework that applies to many modalities and supports early and late fusion scenarios. In every iteration, M-SAM in three steps optimizes learning. **First, it identifies the dominant modality** based on modalities' contribution in the accuracy using Shapley. **Second, it decomposes the loss landscape**, or in another language, it modulates the loss to prioritize the robustness of the model in favor of the dominant modality, and **third, M-SAM updates the weights** by backpropagation of modulated gradients. This ensures robust learning for the dominant modality while enhancing contributions from others, allowing the model to explore and exploit complementary features that strengthen overall performance. Extensive experiments on four diverse datasets show that M-SAM outperforms the latest state-of-the-art optimization and gradient manipulation methods and significantly balances and improves multimodal learning.

## 1 Introduction

Multimodal learning [27] has become the cornerstone of emerging deep neural models. These advanced models integrate diverse data types, including text, audio, images, and sensor data, mimicking the multiplicity of sensory information that humans experience.

This capability for integration is crucial for a range of complex tasks. This integration capability is crucial for a range of complex tasks, enabling models to fully grasp context through multiple viewpoints, essential in a wide spectrum of applications such as autonomous driving, human-computer interaction, and intelligent virtual agents. By blending various sources of information, these multi-modal approaches can construct richer and more robust representations, potentially leading to improved performance across various domains. However, achieving the ideal balance among modalities presents a

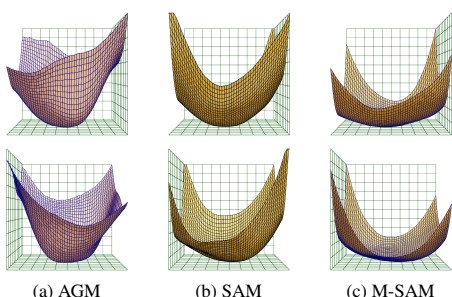

(a) AGM  (b) SAM  (c) M-SAM

Figure 1: Loss landscape visualization of CREMA-D (late fusion) for AGM, SAM, and M-SAM from two different viewpoints.

39th Conference on Neural Information Processing Systems (NeurIPS 2025).

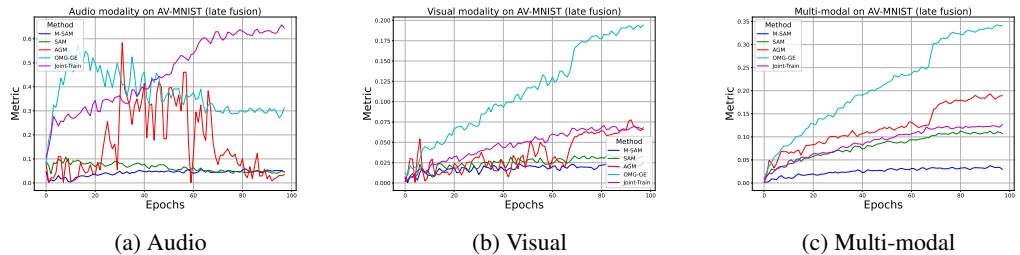

|     |     |     |
| --- | --- | --- |
| (a) Audio | (b) Visual | (c) Multi-modal |

Figure 2: The normalized overfitting gap ($\tau$) comparisons of joint-trained, OGM-GE, AGM, SAM, and our proposed M-SAM on the AV-MNIST dataset. Viewing this in color is recommended.

unique challenge, as one modality can dominate the learning process, limiting the model's ability to develop unique discriminative capabilities from less dominant data sources. In practice, the imbalanced modality can lead to unexpected results, where the performance of jointly trained models shows minimal improvement or even declines compared to unimodal models. [4, 33]. Some studies have found that different modalities often converge at varying speeds, a phenomenon known as uncoordinated convergence [23, 28, 30, 34, 16]. Modality encoders sometimes overfit due to the nature of the input data and the underlying task, especially when overexposed to their training data.[23, 28, 6]. This overexposure causes the model to adhere too closely to the noise and unique characteristics of the training data, resulting in poor generalization to new, unseen data. Similarly, the shared network, which integrates features from all modalities, can be overfitted to the combined feature space, reducing its ability to generalize effectively. This uncoordinated convergence can significantly hinder the performance of multi-modal models.

While many researchers [28, 23] argue that multi-modal networks fail to fully utilize individual modalities if their performance drops in mono-modality evaluations compared to single-modality training, such claims may overlook key aspects of how these networks are structured. For instance, late fusion techniques [41, 38, 8], where modalities are kept separate until the final layers and allow each modality's impact to be more easily assessed. In contrast, early fusion merges modalities from the start and process them together. In these cases, the contributions of individual modalities become intertwined, making their individual effects less distinguishable but not necessarily less significant. The fact that a modality does not perform as well individually within a complex multi-modal system does not mean it is not providing valuable information. Evaluating these networks requires a more profound look rather than a straightforward comparison of isolated performances.

Recent SOTA [23, 28] studies, which utilize gradient modulation techniques, tailor backpropagation for each modality, aiming to find minima that are perfectly suited to each modality. While this approach does outperform previous modulation-based works by achieving better accuracy on both modalities and overall, huge gap shows it tends to overfit [18], leading to sharper minima [12] that constrain the model's ability to explore shared and complementary high semantic cues across modalities. The degree of overfitting is evaluated using the normalized overfitting gap, $\tau$, measuring the discrepancy between a model's training and test performance. It is defined as:

$$\tau = \frac{|Acc_{train} - Acc_{test}|}{Acc_{test}}, \tag{1}$$

with $0 \leq \tau \leq 1$, and assuming models are not underfitted, values closer to 0 signify robust generalization, whereas values nearing 1 indicate pronounced overfitting. Such a normalized overfitting gap has been established as a proxy for sharpness and generalization ability [18, 7], and it clearly illustrates the advantage of our method in learning flatter, generalizable minima. Figure 2 shows that throughout training, our M-SAM method maintains a close match between training and validation accuracy, indicating a well-balanced learning process. Figure 1 further highlights differences in curvature by visualizing the loss landscapes. The relationship between flatter, wider minima and reduced overfitting gap is clearly reflected in M-SAM. By relaxing the strict perturbation constraint used in traditional SAM, M-SAM finds broader, flatter minima. This outcome aligns with the findings of Friendly-SAM [24] that argues traditional SAM not necessarily gets to the flattest minima.

In our paper, we present Modality-Aware SAM (M-SAM), a method designed to boost network generalization by giving precedence to the dominant modality. By steering the learning process towards flatter minima that favor the dominant modality, M-SAM enhances its resilience against parameter updates from other modalities during backpropagation. Additionally, this approach allows non-dominant modalities the freedom to explore and search for optimal features aligned with overall performance. Unlike recent studies that have only slightly improved encoder performance through narrow-focused gradient modulation techniques, M-SAM actively searches for broader, flatter areas in the loss landscape that benefit all modalities and the shared network. This approach prevents our system from becoming overly fine-tuned to the particular characteristics of the training data. The proposed method also dynamically adjusts the learning process by continuously evaluating and rebalancing the contribution of each modality in favor of the dominant modality's generalization in each iteration that not only enhances the weaker modalities to search for their optima but also ensures a more balanced, robust learning across the board. As our main contributions, we show that:

- Our method is conceptually aligned with findings that dominant modalities can emerge during multi-modal training due to dynamics such as modality competition [15]. We propose a practical optimization-based solution that dynamically adapts to such imbalances during training. Specifically, we introduce Modality-Aware SAM (M-SAM), which detects and prioritizes the dominant modality at each iteration and mini-batch, ensuring its robustness against noisy gradient contributions from weaker modalities and promoting more stable multi-modal convergence.

- M-SAM with a modality-focused generalization capability enhances the resilience and stability of the dominant modality against weight updates from other modalities while simultaneously preventing underfitting phenomena in non-dominant modalities.

- Using the architecture proposed by [23] as a baseline, we evaluated M-SAM across diverse datasets and scenarios, including early and late fusion. Our experiments demonstrate consistent and substantial performance improvements over state-of-the-art methods in optimizer design, highlighting the versatility and effectiveness of M-SAM.

## 2 Related works and backgrounds

### 2.1 Multi-modal learning

Multi-modal learning is a rapidly growing field in research, aiming to efficiently process data from multiple senses for practical applications. It spans various domains, including multi-modal recognition [37] and understanding audio-visual scenes [43]. Researchers have highlighted the advantages of multi-modal learning over uni-modal approaches [14], while others have delved into the challenges and failures associated with this type of learning [15]. Despite efforts to enhance performance with more information, studies [5, 30, 34, 36] have shown that many multi-modal learning methods struggle due to discrepancies between modalities. To address these challenges, researchers have proposed innovative approaches such as OGM-GE [28], which adaptively controls optimization by monitoring modality contributions. G-Blending [34] computes optimal blending based on modality behaviors, while MSES [8] detects overfitting and performs early stopping. MSLR [38] effectively builds late-fusion multi-modal models, and AGM [23] boosts performance with adaptive gradient modulation and fusion strategies. Kontras et al. [20] considered uni-modal loss values beside multimodal-gradient to modulate encoder gradients, and [35] customized Pareto approach for multimodal scenarios. These advancements aim to improve the effectiveness of multi-modal learning models.

### 2.2 SAM

Sharpness-Aware Minimization (SAM) was first proposed by [7] as a machine learning technique to enhance model generalization. It achieves this by concurrently minimizing the loss value and the sharpness of the loss function, aiming for flatter minima. SAM has demonstrated efficacy beyond deep neural networks, finding applications in diverse fields such as language models [1] and fluid dynamics [17], illustrating its adaptability. Adaptive SAM (ASAM) [21] improves SAM by dynamically adjusting the sharpness radius, addressing the parameter scale-dependency issue. Surrogate Gap Guided Sharpness-Aware Minimization (GSAM) [44] introduces the surrogate gap, simplifying sharpness measurement in local minima. Fisher SAM [19] enhances SAM by using ellipsoids from Fisher information, refining neighborhood structures for better sharpness estimation. GAM [40] presents first-order flatness focusing on the maximal gradient norm within a perturbation

radius which bounds both the maximal eigenvalue of Hessian at local minima and the regularization function of SAM. SAM finds applications in various machine-learning tasks [3, 26, 1, 39]. Chen et al. [3] use SAM on Vision Transformer and MLP-Mixers to enhance accuracy and robustness. Na et al. [26] note SAM's role in parameter compressibility and task transfer. Yeo et al. [39] apply SAM to medical imaging, improving segmentation while balancing sharpness and warp regularization. However, SAM has not been used in multi-modal learning so far.

# 3 Multi-modality learning and SAM

## 3.1 Notation and Problem Definition

In tasks where we deal with datasets composed of multiple modalities, the training set $D = \bigcup_{i=1}^{N} \{\bigcup_{m=1}^{M} (x_m^i, y^i)\}\}$ often includes a variety of data types, where $m \in [1, \ldots, M]$ indicates the modality to which the data sample $(x_m^i, y^i)$ belongs. These modalities could represent different sources or types of data, like images, audio, texts, and sensor readings. To address the imbalance gradients across these modalities, we partition the whole set as $D = \bigcup_{m=1}^{M} (X_m, Y)$, which $X_m$ and $Y$ contain data from $m$th modality and it's corresponding label and $N$ represents training samples population.

## 3.2 Preliminaries

Our objective is to train a system that can accurately acquire and process data from all modalities, ensuring that each contributes to the back-propagation considering the optimization stage, despite their differences in representation. From $m$th modality's perspective, the system, modeled by a function $f_m(\cdot; \theta_m)$ parameterized by $\theta_m$, aims to predict the correct label $\hat{y}^i$ for a given sample $x^i$. The loss function, among other variants, acts as the guide $\ell(\hat{y}^i, y^i)$ to fine-tune the parameters $\theta_m$ during training. The training task would be formulated as:

$$\theta^* = \min_\theta \sum_{i=1}^{N} \mathcal{L}\bigg(f^s\big(f_1^e(x_1^i \cdots x_M^i; \theta_1) \oplus_1 f_2^e(x_1^i \cdots x_M^i; \theta_2) \cdots \oplus_{M-1} f_M^e(x_1^i \cdots x_M^i; \theta_M)\big), y^i\bigg),$$
(2)

where $\oplus_i$ denotes an arbitrary fusion operation, such as early, late, or any hybrid fusion methods and $f_m^e$ represents the encoder associated with $m$th modality. $\theta^*$ is the optimal parameter set we seek, defined such that $\{\theta_i \subset \theta \mid \forall 1 \leq j \leq M; \theta_i \cap \theta_j \neq \varnothing\}$. Note that, $\theta_m$ represents a portion of architecture parameters including neural connections, activation function, etc. that carry out the effect of $x_m^i$ to the output. Being more precise, $\theta_m := \{\theta_m^e \cup \theta_{mk}^e \mid k \in \{1, 2, \ldots, M\}, m \neq k\}$, where $\theta_m^e$ represents the modality-specific encoder architecture parameters of $m$th modality. $\theta_{mk}^e$ defines the model's parameters associated with the fusion network that fuses modality $k$th to modality $m$th. The shared network $f^s$ and its parameters $\theta^s$ in $f^s(., \theta^s)$ are influenced by forward/backward propagation of various modalities. Considering these definitions, Eq. 2, without loss of generalization can be comprehended as follows;

$$\theta^* = \min_\theta \sum_{i=1}^{N} \mathcal{L}\big(f(x_1^i, x_2^i, \ldots, x_M^i; \theta), y^i\big) = \min_\theta \sum_{i=1}^{N} \mathcal{L}\big(f^s(\phi_1, \phi_2, \ldots, \phi_M; \theta^s), y^i\big), \quad (3)$$

where $f$ and $\theta$ in $f(., \theta)$ respectively are the model and its parameters. On the other hand, $\phi_m = f_m^e(x_1^i, \ldots, x_m^i, \ldots, x_M^i; \theta_m^e, \theta_{m1}^s, \theta_{m2}^s, \ldots, \theta_{mk}^s \mid_{m \neq k}, \ldots, \theta_{mM}^s)$ is defined as the feature representation produced by the $m$-th encoder taking into account other modalities' effect through the fusion network's parameters, $\theta_{mj}$s.

Considering Eq. 3, the loss function of the multi-modal underlying task can be expressed as the sum of the losses from all modalities, as $L = \frac{1}{N} \sum_{i=1}^{N} \mathcal{L}\big(f^s(\phi_1, \phi_2, \ldots, \phi_M; \theta^s), y^i\big)$. The intricate interplay among modalities during model training, coupled with the complexity of deep neural network architectures, poses significant challenges in isolating the individual contributions of each modality to overall performance. Utilizing the Shapley approach as outlined by [29], offers a robust mechanism to decompose the contributions of each modality to the output loss on a per mini-batch basis (Appendix. B)

$$L = (v_1 + v_2 + \cdots + v_M)L = L_1 + L_2 + \cdots + L_M, \quad (4)$$

where $v_m$ represents contribution of the $m$th modality in the output. Note that $\sum_M v_m = 1$.

### 3.3 Learning Dynamics and Loss Landscape

Let $\psi = f^s(\phi_1, \phi_2, \ldots, \phi_m, \ldots, \phi_M; \theta^s)$ denote the combined feature representation that integrates the contributions from all modalities, and $\theta^s$ represents shared weights associated to the downstream task. Given that, the update rules for the parameters are as follows:

$$\theta^s_{m_{(t+1)}} = \theta^s_{m_{(t)}} - \eta \nabla_{\theta^s} L(\theta^s_{m_{(t)}}) = \theta^s_{m_{(t)}} - \eta \frac{1}{N} \sum_{i=1}^{N} \left( \frac{\partial \mathcal{L}(\psi, y^i)}{\partial \psi} \frac{\partial \psi}{\partial \theta^s} \right), \tag{5}$$

$$\theta^e_{m_{(t+1)}} = \theta^e_{m_{(t)}} - \eta \nabla_{\theta^e} \mathcal{L}(\theta^e_{m_{(t)}}) = \theta^e_{m_{(t)}} - \eta \frac{1}{N} \sum_{i=1}^{N} \left( \frac{\partial \mathcal{L}(\psi, y^i)}{\partial \psi} \frac{\partial \psi}{\partial \phi_m} \frac{\partial \phi_m}{\partial \theta^e_m} \right), \tag{6}$$

where $\eta$ is the learning rate, $\mathcal{L}$ is the loss function, and $t$ indexes the iteration of the gradient descent. In Eq. 5, $\frac{\partial \psi}{\partial \theta^s}$ represents the component where inter-modality interaction occurs. During backpropagation, this term adjusts the shared network weights, $\theta^s$, based on the gradients contributed by all modalities. Considering the decomposition of the loss landscape described in Eq. 4 and depicted in Fig. 3, as training progresses, if the dominant modality, $m_0$, is guided toward a wide, flat minimum in the loss landscape, this ensures that even when inter-modality interactions slightly deviate the shared-network parameters, $\theta^s$ and move the dominant modality from its optimal minima, it remains within the bounds of a wide, stable minimum. Consequently, weight updates in the shared parameter space, $\theta^s$, may shift the network parameters slightly. Still, the dominant modality's contribution to the overall loss remains relatively small compared to the case where it converges to a sharp minimum.

### 3.4 SAM

SAM is an optimization technique that improves generalization by finding flatter minima by minimizing the maximum loss in a neighborhood around the parameters. Specifically, consider a family of models parameterized by $\theta \in \mathcal{W} \subseteq \mathbb{R}^d$; $\mathcal{L}$ is the loss function over the training dataset $\mathcal{D}$. SAM aims to minimize the following upper bound of the Probably Approximately Correct (PAC)-Bayesian generalization error for any $\rho > 0$,

$$\mathcal{L}(\theta) \leq \max_{\|\epsilon\|_p \leq \rho} \mathcal{L}(\theta + \epsilon) + \frac{\lambda}{2}\|\theta\|^2. \tag{7}$$

To solve the above minimax problem, at each iteration $t$, SAM updates

$$\begin{aligned} \epsilon_t &= \frac{\rho \cdot \text{sign}(\nabla \mathcal{L}(\theta_{t-1})) |\nabla \mathcal{L}(\theta_{t-1})|^{q-1}}{\left( \|\nabla \mathcal{L}(\theta_{t-1})\|_q^q \right)^{1/p}}, \\ \theta_t &= \theta_{t-1} - \eta_t \left( \nabla \mathcal{L}(\theta_{t-1} + \epsilon_t) + \lambda \theta_{t-1} \right), \end{aligned} \tag{8}$$

Here $1/p + 1/q = 1$, $\rho > 0$ is a hyperparameter, $\lambda > 0$ is the parameter for weight decay, and $\eta_t > 0$ is the learning rate. By setting $p = q = 2$ and introducing an intermediate variable $\mathbf{u}_t$, we have:

$$u_t = \theta_{t-1} + \frac{\rho \nabla \mathcal{L}(\theta_{t-1})}{\|\nabla \mathcal{L}(\theta_{t-1})\|} = \theta_{t-1} + \epsilon_t, \tag{9}$$

$$\theta_t = \theta_{t-1} - \eta_t \left( \nabla \mathcal{L}(u_t) + \lambda \theta_{t-1} \right). \tag{10}$$

### 3.5 Modality-Aware SAM

To enhance the overall performance of models on multimodal datasets, the SAM optimization process can be adapted by incorporating a split loss function (Eq. 4). This reformulation targets the dominant modality, making it more robust against changes in network parameters due to backpropagation from other modalities. While SAM's modality-agnostic nature originally confines its generalization capability, this new approach ensures that the proposed M-SAM provides strong generalization power in the context of diverse and complex multimodal data. Utilizing Eq.7 and Eq.4, we can reformulate SAM as follows:

$$\min_{\theta} \max_{\|\epsilon\|_p \leq \rho} [L_1(\theta + \epsilon) + \cdots + L_M(\theta + \epsilon)] + \frac{\lambda}{2}\|\theta\|^2. \tag{11}$$

In this regard, the perturbation defined in Eq. 9 can also be divided into components associated with each modality as $\epsilon_t = \epsilon_t^1 + \epsilon_t^2 + \cdots + \epsilon_t^{m_0} + \cdots + \epsilon_t^M$, where $m_0$ is the dominant modality. When

---
**Algorithm 1** M-SAM Algorithm
---
**Require:** Training dataset $\mathcal{D} = \bigcup_{i=1}^{N} \{ \bigcup_{m=1}^{M} (x_m^i, y^i) \} \}$, neural network $f(\cdot)$ with parameters $\theta$, loss function $\mathcal{L}$, mini-batch size $b$, learning rate $\eta$, neighborhood size $\rho$, weight decay coefficient $\lambda$,
**Ensure:** Trained parameters $\theta^*$
1: Initialize parameters $\theta_0$, $t = 0$
2: **while** not converged **do**
3:      Sample minibatch $\mathcal{B} = \{((x_1^1, \ldots, x_M^1), y^1), \ldots, ((x_1^b, \ldots, x_M^b), y^b)\}$
4:      $\mathcal{L}(\theta_t) = \sum_M \mathcal{L}(f(x_m^i; \theta_t), y^i) = \sum_M v_m \mathcal{L}(f(x_1^i, \ldots, x_M^i; \theta_t), y^i)$      $\triangleright$ Eq. 4
5:      $\mathcal{L}_d(\theta_t) = v_d \mathcal{L}(f(x_1^i, \ldots, x_M^i; \theta_t), y^i) \mid m_d = \arg\max\limits_{m \in \{1,\ldots,M\}} v_m$
6:      $\mathcal{L}_s(\theta_t) = \sum\limits_m v_m \mathcal{L}(f(x_1^i, \ldots, x_M^i; \theta_t), y^i)$ , $\forall m \in \mathcal{M} = \{m \in \{1, \ldots, M\}, m \neq m_d\}$
7:      $\nabla \mathcal{L}_d(\theta_t) = \text{Backward}(\mathcal{L}_d, f(\cdot))$,   $\nabla \mathcal{L}_s(\theta_t) = \text{Backward}(\mathcal{L}_s, f(\cdot))$
8:      $\epsilon_t^d = \rho \frac{\nabla \mathcal{L}_d(\theta_t)}{\|\nabla \mathcal{L}_d(\theta_t)\|_2}$
9:      $\theta_t \leftarrow \theta_t - \eta_t \left[ \nabla \mathcal{L}_d(\theta_t + \epsilon_t^d) + \nabla \mathcal{L}_s(\theta_t) + \lambda \theta_t \right]$
10:      $t \leftarrow t + 1$
11: **end while**
---

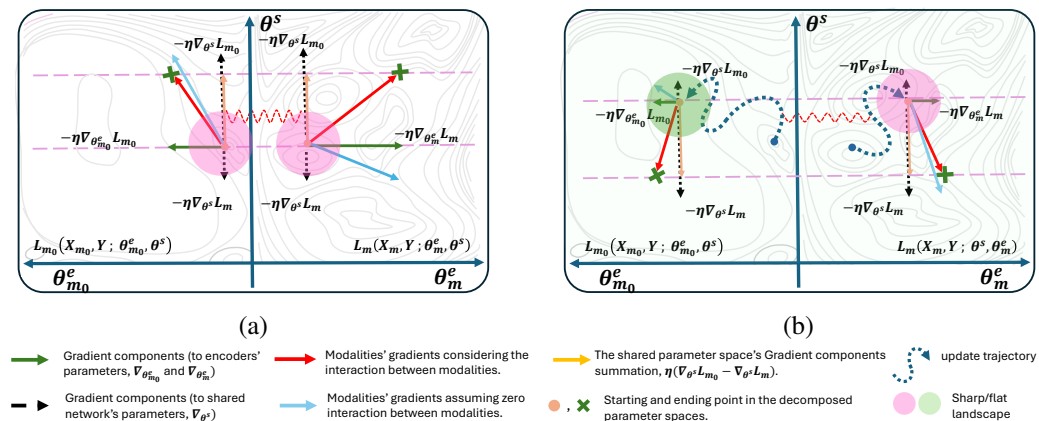

(a)             (b)

Figure 3: **Illustration of the interaction of gradients in sharp and flat loss landscapes (before and after convergence of MSAM).** In (a), the sharp landscape amplifies the dominant modality $m_0$'s gradient magnitude, often overshadowing the non-dominant modality due to large and differing gradient directions in the shared parameter space. In (b), settling the dominant modality $m_0$ in a flatter minimum reduces its gradient magnitude across a wide range of parameter neighborhoods, allowing the non-dominant modality to explore the parameter space more freely and promoting balanced optimization across modalities. Note that the loss landscape decomposition is achieved using the Shapley method by dividing total loss to modality-specific loss in every iteration.

the dominant modality approaches or settles on a minimum, its contribution to the loss function during perturbations is significantly greater than that of other modalities. Therefore, making this dominant modality robust against perturbations from other modalities allows the latter to search for their minima without substantially affecting the dominant modality or the overall performance of the model. However, there is no guarantee that $\epsilon_t^{m_0} >> \epsilon_t^m, \forall m \in \{1, 2, \ldots, M \mid m \neq m_0\}$. As a result, the traditional SAM can not provide focused generalization in favor of the dominant modality. Inspired by [42], by ignoring the SAM optimization term of non-dominant modalities in Eq. 11, our proposed modality-aware SAM optimization target is achieved as

$$\min_\theta \overbrace{\left[ \max_{\|\epsilon\|_p \leq \rho} L_{m_0}(\theta + \epsilon_{m_0}) - L_{m_0}(\theta) \right]}^{\text{Optimization term for the dominant modality}} + L_{m_0}(\theta) + \overbrace{[L_1(\theta) + L_2(\theta) + \cdots + L_M(\theta)]}^{\text{optimization term for non-dominant modalities}} + \frac{\lambda}{2} \|\theta\|^2.$$

The gradient update equation, Eq. 10, is modified as follows;

$$\theta_t = \theta_{t-1} - \eta_t \Big( \nabla L_{m_0} \left( \theta_{t-1} + \epsilon_t^{m_0} \right) + \nabla L_1 \left( \theta_{t-1} \right)$$
$$+ \nabla L_2 \left( \theta_{t-1} \right) + \cdots + \nabla L_M \left( \theta_{t-1} \right) + \lambda \theta_{t-1} \Big), \tag{12}$$

where $\epsilon_t^{m_0}$ calculated as $\epsilon_t^{m_0} = \frac{\rho \nabla L_{m_0}(\theta_{t-1})}{\|\nabla L_{m_0}(\theta_{t-1})\|}$.

As illustrated in Eq. 12, the proposed modality-aware SAM bypasses sharpness-aware minimization for non-dominant inputs, which are less prone to overfitting due to being overshadowed by the dominant modality. We adopted the Shapely value proposed by [23], elaborated in Appendix. B to split the loss into individual modalities' loss. Algorithm. 1 outlines the complete M-SAM utilizing SGD as the base gradient optimizer.

## 3.6 Stability and Convergence Analysis of M-SAM

Following the analytical framework in [22], we establish the convergence behavior of M-SAM under its dynamic update rule and modality-aware selection mechanism. For analytical tractability and without loss of generality, we make the following assumptions:

1. $\mathcal{L}(\theta_t) = \sum_{m=1}^{M} \nu_m \mathcal{L}_m(\theta_t)$ and its gradient is bounded $K$-smooth ($K$-Lipschitz), i.e., $\|\nabla \mathcal{L}(\theta_t) - \nabla \mathcal{L}(\hat{\theta}_t)\| \leq K\|\theta_t - \hat{\theta}_t\|, \forall \theta_t, \hat{\theta}_t$, that $\|\nabla \mathcal{L}(\theta_t)\| \leq G_{\max}$.

2. Learning rate $\eta_t = \frac{\eta_0}{\sqrt{t}}$ and perturbation $\rho_t = \frac{\rho_0}{\sqrt{t}}$. for analytical tractability. In theory, this smooth decay schedule facilitates convergence analysis. In practice, we adopt a step-wise decay, by multiplying the learning rate by 0.1 every 70 iterations, which decays $\eta_t$ even faster. This empirical schedule still satisfies the convergence conditions and does not compromise the theoretical guarantees.

considering $d_t = \theta_{t+1} - \theta_t = -\eta_t \nabla \mathcal{L}(\hat{\theta}_t)$ and $\hat{\theta}_t = \theta_t + \rho_t \frac{\nabla_{m_0} \mathcal{L}(\theta_t)}{\|\nabla_{m_0} \mathcal{L}(\theta_t)\|}$ where $\nabla_{m_0}$ means gradient toward the dominant modlaity. Using descent Lemma for a $k$-smooth function we can write:

$$\mathcal{L}(\theta_{t+1}) - \mathcal{L}(\theta_t) \leq -\eta_t \langle \nabla \mathcal{L}(\theta_t), \nabla \mathcal{L}(\hat{\theta}_t) \rangle + \frac{K\eta_t^2}{2} \|\nabla \mathcal{L}(\hat{\theta}_t)\|^2 =$$
$$- \eta_t \langle \nabla \mathcal{L}(\theta_t), \nabla \mathcal{L}(\theta_t) - \nabla \mathcal{L}(\theta_t) + \nabla \mathcal{L}(\hat{\theta}_t) \rangle + \frac{K\eta_t^2}{2} \|\nabla \mathcal{L}(\hat{\theta}_t)\|^2 =$$
$$- \eta_t \|\nabla \mathcal{L}(\theta_t)\|^2 - \eta_t \langle \nabla \mathcal{L}(\theta_t), \nabla \mathcal{L}(\hat{\theta}_t) - \nabla \mathcal{L}(\theta_t) \rangle + \frac{K\eta_t^2}{2} (\|\nabla \mathcal{L}(\hat{\theta}_t)\|^2) \leq \tag{13}$$
$$- \eta_t \|\nabla \mathcal{L}(\theta_t)\|^2 + \eta_t \langle \nabla \mathcal{L}(\theta_t), \nabla \mathcal{L}(\theta_t) - \nabla \mathcal{L}(\hat{\theta}_t) \rangle + K\eta_t^2 G_{max}^2 \leq$$
$$- \eta_t \|\nabla \mathcal{L}(\theta_t)\|^2 + \eta_t \|\nabla \mathcal{L}(\theta_t)\| \|\nabla \mathcal{L}(\theta_t) - \nabla \mathcal{L}(\hat{\theta}_t)\| + K\eta_t^2 G_{max}^2 \leq$$
$$- \eta_t \|\nabla \mathcal{L}(\theta_t)\|^2 + K\eta_t \rho_t^2 G_{max} + K\eta_t^2 G_{max}^2$$

Although the upper-bound term appears to capture potential instability introduced by dynamic modality switching, it can be shown that the bound is a function of $\rho_t$ alone. Hence, the convergence rate of M-SAM remains invariant to the gradient of the selected modality $\nabla_{m_0} \mathcal{L}(\theta_t)$, ensuring that modality selection does not affect stability.

by rearranging the inequality, it would be as:

$$\eta_t \|\nabla \mathcal{L}(\theta_t)\|^2 \leq \mathcal{L}(\theta_t) - \mathcal{L}(\theta_{t+1}) + K\eta_t \rho_t^2 G_{max} + K\eta_t^2 G_{max}^2 \tag{14}$$

now do summation over all iterations, $1 \leq t \leq T$. Using telescope sum properties for loss terms on the right side of the inequality and this property, $\frac{\eta_0}{\sqrt{T}} \sum_{t=1}^{T} \|\nabla \mathcal{L}(\theta_t)\|^2 \leq \sum_{t=1}^{T} \eta_t \|\nabla \mathcal{L}(\theta_t)\|^2$:

$$\frac{\eta_0}{\sqrt{T}} \sum_{t=1}^{T} \|\nabla \mathcal{L}(\theta_t)\|^2 \leq \mathcal{L}_1(\theta_t) + K\eta_0 \rho_0^2 G_{max} \sum_{t=1}^{T} \frac{1}{t\sqrt{t}} + K\eta_0^2 G_{max}^2 \sum_{t=1}^{T} \frac{1}{t} \tag{15}$$

Considering that $\sum_{t=1}^{T} \frac{1}{t} \leq 1 + \log T$ and $\sum_{t=1}^{T} \frac{1}{t^{3/2}}$ forms a $p$-series with $p = \frac{3}{2} > 1$, whose partial sum remains bounded, the only term on the right-hand side influencing the asymptotic behavior is the harmonic component. Therefore, the overall convergence rate of M-SAM is $\mathcal{O}\left(\frac{\log T}{\sqrt{T}}\right)$.

$$\frac{1}{T} \sum_{t=1}^{T} \|\nabla \mathcal{L}(\theta_t)\|^2 \leq \mathcal{O}\left(\frac{\log T}{\sqrt{T}}\right).$$

That means, As $T$ increases, the average gradient decreases at rate $\frac{\log T}{\sqrt{T}}$. This is similar to what we get for methods like SGD and traditional SAM.

## 4 Experiments and Discussion

### 4.1 Dataset

M-SAM is evaluated on three popular multi-modal datasets: AV-MNIST [33], CREMA-D [2], UR-Funny [11], and AVE [31]. Details of these datasets are presented in Appendix. A

### 4.2 Networks' architecture and preprocessing

To evaluate our approach's performance in both late and early fusion strategies across various multi-modal datasets, we followed a unified model design based on prior works by [23, 25]. For early fusion, we used the MAXOUT network [9] for the fusion module. Consistent encoder architectures were maintained for each dataset in both fusion strategies. Specifically, ResNet18 was used as the encoder for the audio and visual modalities in the AV-MNIST and CREMA-D datasets. In contrast, the UR-Funny dataset utilized a Transformer [32] encoder for all three modalities. To ensure consistency with previous works, we followed the preprocessing steps they applied. For CREMA-D, we extracted 1 frame per minute (fpm) from each clip and processed the audio data into a spectrogram of size $257 \times 299$ with a window length of 512 and an overlap of 353. We used SGD with 0.9 momentum and $10^{-4}$ weight decay as the optimizer. The learning rate was initially set to $10^{-3}$ and was multiplied by 0.1 every 70 epochs. For UR-Funny, we utilize the preprocessed data introduced by [25].

### 4.3 Results and discussion

To show the effectiveness of M-SAM, we compared it with mainstream multi-modal optimizer enhancement approaches: MSES [8], MSLR [38], OGM-GE [28], AGM [23], MM-Pareto [35], CGGM [10], and Recon-Boost [13]. Our findings reveal that M-SAM consistently outperforms all other methods in overall performance and mono-modal accuracy across most scenarios. However, we believe its final performance metrics do not solely determine a method's efficacy; the trends observed throughout the training epochs also provide valuable insights. Therefore, we analyzed the accuracy of the validation set over these epochs. These accuracy curves highlight how the method handles various modalities, which can sometimes conflict during training. The smooth and steady progression of M-SAM's overall accuracy curves demonstrates its robustness in managing the complexities of multi-modal learning and its ability to harmonize different modalities effectively.

#### 4.3.1 Early Fusion Setup

In early fusion, Table. 2, modality-specific representations are combined before classification, which causes their gradients to interact directly in the shared feature space. Joint-Train performs poorly in this setup, particularly on CREMA-D and UR-Funny, where modality imbalance is more severe. AGM and MM-Pareto provide modest improvements, with MM-Pareto benefiting from its multi-objective loss formulation. CGGM and Recon-Boost also perform competitively. CGGM explicitly supports early fusion by computing modality-specific gradients through separate forward passes, even when using a shared classifier. It modulates training based on the difference between each modality's gradient and the joint gradient, without requiring modality-specific heads. Recon-Boost, while evaluated primarily with decision-level fusion, still offers competitive performance under early fusion. Its alternating update strategy and reconcilement loss help manage imbalance, though its training dynamics are better suited to architectures with modality-specific learners.

Table 1: Accuracy ($Acc.$), and single-modal accuracy ($Acc_a, Acc_v, Acc_t$) on the AV-MNIST, CREMA-D, UR-Funny, and AVE datasets using **late fusion** architecture. Please note that the OGM-GE method could not extend to more than two modality cases in their original shape.

| Model | AV-MNIST [33] | | | CREMA-D [2] | | | UR-Funny [11] | | | | AVE [31] | | |
|---|---|---|---|---|---|---|---|---|---|---|---|---|---|
| | $Acc_a$ | $Acc_v$ | $Acc_{mm}$ | $Acc_a$ | $Acc_v$ | $Acc_{mm}$ | $Acc_a$ | $Acc_v$ | $Acc_t$ | $Acc_{mm}$ | $Acc_a$ | $Acc_v$ | $Acc_{mm}$ |
| Single-audio | 39.61 | .. | .. | 52.12 | .. | .. | 59.23 | .. | .. | .. | 65.43 | .. | .. |
| Single video | .. | 65.14 | .. | .. | 60.37 | .. | .. | 53.16 | .. | .. | .. | 64.58 | .. |
| Single-text | .. | .. | .. | .. | .. | .. | .. | .. | 63.46 | .. | .. | .. | .. |
| Joint-Train | 14.59 | 62.85 | 68.41 | 56.08 | 44.32 | 61.19 | 50.31 | 53.51 | 49.78 | 64.50 | 59.10 | 63.72 | 73.19 |
| MSES [8] | 27.50 | 63.34 | 70.68 | 55.31 | 45.72 | 64.13 | 55.31 | 49.69 | 57.87 | 64.23 | 65.84 | 71.93 | 76.47 |
| MSLR [38] | 22.72 | 62.92 | 70.62 | 55.75 | 47.84 | 62.93 | 53.14 | **53.59** | 46.93 | 65.52 | 70.39 | 69.41 | 75.22 |
| OGM-GE [28] | 24.53 | 55.85 | 71.08 | 58.15 | **58.90** | 64.42 | .. | .. | .. | .. | 67.91 | 71.09 | 75.53 |
| AGM [23] | 38.90 | 63.65 | 72.14 | 56.35 | 54.12 | 64.72 | 54.87 | 49.36 | 62.22 | 65.97 | 70.68 | 72.34 | 77.11 |
| MM-Pareto [35] | 42.17 | 64.31 | 73.22 | 61.85 | 56.94 | 66.63 | 54.59 | 52.12 | 62.37 | 67.04 | 70.31 | **73.88** | 77.68 |
| CGGM [10] | 39.53 | 64.13 | 73.42 | 57.14 | 55.07 | 67.03 | **55.21** | 49.83 | 62.74 | 67.43 | 70.91 | 73.17 | 77.83 |
| Recon-Boost [13] | 40.12 | 64.18 | 73.59 | 57.08 | 54.83 | 67.47 | 55.13 | 50.08 | **62.88** | 67.61 | **71.13** | 73.48 | 78.02 |
| SAM | 36.31 | 64.67 | 73.17 | 60.69 | 51.43 | 66.32 | 53.67 | 51.37 | 61.48 | 65.95 | 66.13 | 72.83 | 77.66 |
| M-SAM | **41.93** | **64.97** | **74.08** | **62.78** | 53.22 | **68.56** | 51.56 | 52.67 | 60.17 | **68.31** | 68.27 | 72.57 | **79.67** |

Table 2: Accuracy ($Acc.$), and single-modal accuracy ($Acc_a, Acc_v, Acc_t$) on the AV-MNIST, CREMA-D, UR-Funny, and AVE datasets using **early fusion** architecture.

| Model | AV-MNIST [33] | | | CREMA-D [2] | | | UR-Funny [11] | | | | AVE [31] | | |
|---|---|---|---|---|---|---|---|---|---|---|---|---|---|
| | $Acc_a$ | $Acc_v$ | $Acc_{mm}$ | $Acc_a$ | $Acc_v$ | $Acc_{mm}$ | $Acc_a$ | $Acc_v$ | $Acc_t$ | $Acc_{mm}$ | $Acc_a$ | $Acc_v$ | $Acc_{mm}$ |
| Joint-Train | 24.28 | 60.14 | 71.15 | 55.31 | 51.72 | 62.13 | 54.87 | 50.86 | 54.14 | 65.15 | 67.40 | 71.85 | 76.29 |
| AGM [23] | **47.79** | **68.48** | 72.26 | 51.42 | 47.54 | 64.09 | **64.88** | **55.20** | 63.36 | 66.07 | 68.85 | 72.46 | 77.08 |
| MM-Pareto [35] | 39.82 | 66.15 | 72.74 | 56.90 | 52.83 | 65.30 | 58.32 | 53.08 | 60.45 | 65.92 | 68.52 | 72.18 | 76.83 |
| CGGM [10] | 41.30 | 67.27 | **73.11** | **57.84** | 53.67 | 66.90 | 60.42 | 53.40 | 62.53 | 66.98 | 69.01 | **72.81** | 77.32 |
| Recon-Boost [13] | 40.94 | 67.01 | 72.97 | 57.56 | **54.01** | 66.74 | 59.88 | 53.26 | 62.18 | 66.83 | 68.91 | 72.63 | 77.18 |
| SAM | 38.56 | 64.81 | 73.22 | 56.35 | 53.52 | 66.22 | 54.08 | 49.77 | 63.86 | 66.87 | 68.37 | 72.02 | 76.76 |
| M-SAM (Ours) | 45.63 | 67.72 | **74.48** | 56.83 | 53.71 | **68.43** | 63.20 | 54.77 | **65.24** | 67.92 | **70.11** | 72.46 | **78.23** |

M-SAM significantly outperforms all baselines across datasets in $Acc_{mm}$, with gains of +1.4 % to +2.3 % over the closest competitors, CGGM and Recon-Boost. These improvements are attributable to its optimizer design: M-SAM inherits the generalization benefits of SAM while explicitly prioritizing stable training of the dominant modality. Unlike prior methods that rely on static loss weighting or sequential modality updates, M-SAM adjusts its behavior on a per-iteration basis, mitigating gradient interference without requiring architecture changes.

### 4.3.2 Late Fusion Setup

Late fusion provides a strong evaluation setting for CGGM and Recon-Boost. As demonstrated in Table. 1, CGGM leverages classifier-gradient discrepancy to modulate updates, while Recon-Boost alternates training across modalities and applies a reconcilement loss to improve coordination. Both methods consistently outperform AGM and MM-Pareto in $Acc_{mm}$ across most datasets. However, M-SAM achieves the highest performance overall. Unlike prior methods that rely on architectural separation or sequential training, M-SAM operates entirely at the optimizer level. Its intrinsic design encourages consistently flatter convergence across training iterations, preserving the optimization trajectory of the dominant modality while allowing non-dominant modalities greater freedom to adapt. This balance emerges not from hand-tuned loss weights but from the geometry of the update itself. Notably, the lower single-modality accuracies of M-SAM, despite significantly higher $Acc_{mm}$, suggest that it encourages the learning of features that are not independently discriminative but become informative through cross-modal interaction, enabling more effective integration of complementary modality features.

## 5   Conclusion

In this paper, we introduced M-SAM, an optimizer-level framework that extends Sharpness-Aware Minimization (SAM) to address optimization instability in multi-modal learning. M-SAM identifies the dominant modality per batch and aligns the sharpness-aware update direction with its gradient, enabling stable convergence while preserving learning flexibility for other modalities. This mechanism promotes flatter optima without requiring architectural changes or explicit loss weighting. As

illustrated in Figure 1, M-SAM produces a consistently flatter loss landscape compared to baseline methods such as AGM, improving both generalization and robustness. Empirical results across four multi-modal benchmarks demonstrate that M-SAM outperforms a range of strong baselines, from gradient modulation methods such as AGM and CGGM to stage-wise training strategies like Recon-Boost. These findings highlight the effectiveness of optimization-aware strategies in harmonizing modality contributions and improving training stability in multi-modal settings.

## Acknowledgments and Disclosure of Funding

This material is based upon work supported by the National Aeronautics and Space Administration (NASA) under award number 80NSSC23K1393, the National Science Foundation under Grant Number CNS-2232048, and CCF-2553684.

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

## A    Dataset

To evaluate the effectiveness of our approach, we conducted experiments on three widely used multimodal datasets from the domains of affective computing and multimedia. From the affective computing domain, we utilized the CREMA-D and UR-Funny datasets.

Table 3: The datasets specifications.

| Field of Research | Size | Dataset | Modality | Samples | content |
|---|---|---|---|---|---|
| Affective Computing | L | UR-Funny[11] | {a, v, t} | 16,514 | humor |
|  | M | CREMA-D [2] | {a, v} | 7,442 | emotion |
| Multimedia | M | AV-MNIST[33] | {a, v} | 70,000 | digit |
|  | S | AVE[31] | {a, v} | 4,143 | event detection |

The CREMA-D dataset, curated for speech emotion recognition tasks, includes six emotional labels spanning various speech recordings. The UR-Funny dataset, designed for humor detection, incorporates multimodal cues, including text, visual gestures, and acoustic-prosodic features, providing a rich benchmark for affective computing. In the multimedia domain, we employed the AV-MNIST dataset, which focuses on multimedia classification. This dataset features disturbed images paired with audio signals, offering a challenging setting for evaluating cross-modal learning.

Further details about these datasets, including modality types and task descriptions, are provided in Table. 3.

## B    Mono-Modal Contribution in Total Performance

To split the total loss function into the modality-specific loss function as what Eq. 4 shows, we adopted the Shapley metric proposed by [23]. Consider:

$$\Phi(x) = f^s\big(\phi_1, \phi_2, \ldots, \phi_M; \theta^s\big), \tag{16}$$

where $x = (x_1, \ldots, x_M)$ represents all corresponding $M$ modalities of the input and $\phi_m = f_m^e(x_1^i, \ldots, x_M^i; \theta_m^e, \theta_{m1}^s, \ldots, \theta_{mk}^s \mid_{m \neq k}, \ldots, \theta_{mM}^s)$ is defined as the feature representation produced by the $m$-th encoder taking into account other modalities' effect through the fusion network's parameters, $\theta_{mj}$. Lets $\mathcal{M} := \{m\}_{i=1}^M$ denote the set of all modalities. Zero-padding $0_m$ indicates the absence of modality $m$ features. If $S$ is a subset of $\mathcal{M}$, then $\Phi(S)$ indicates that for $m \in S$, $x_m$ is replaced with $0_m$. The mono-modal response for modality $m$ is then defined as:

$$\Phi_m(x) = \sum_{\substack{S \subseteq \mathcal{M} \setminus \{m\} \\ S \neq \emptyset}} \frac{|S|! \, (k - |S| - 1)!}{k!} V_m(S; \Phi), \tag{17}$$

where $V_m(S; \Phi) = \Phi(S \cup \{m\}) - \Phi(S)$. The empty subset is excluded from the summation to ensure the relation:

$$\Phi(x) = \sum_m \Phi_m(x). \tag{18}$$

For illustration, in the case of two modalities, it would be as follows;

$$\Phi_{m_1}(x) = \frac{1}{2}\left[\Phi(\{x_1, x_2\}) - \Phi(\{0_1, x_2\})\right] + \Phi(\{m_1, 0_{m_2}\}). \tag{19}$$

### B.1    From Mono-Modal Attribute to Loss Landscape Decomposition

The Shapley values in Eq. 17 serve as decomposing weight to compute mono-modal contributions into the overall accuracy. They inherently operate in the domain of performance metrics, not loss

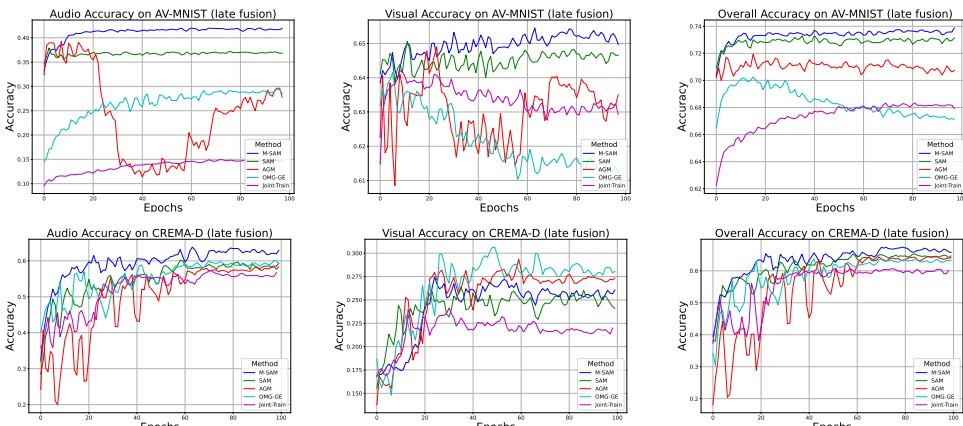

Figure 4: **Performance comparisons of late fusion settings:** Joint-Trained, AGM, OGM-GE, SAM, and our proposed Modality-Aware SAM, on the AV-MNIST and CREMA-D datasets. Viewing this in color is recommended.

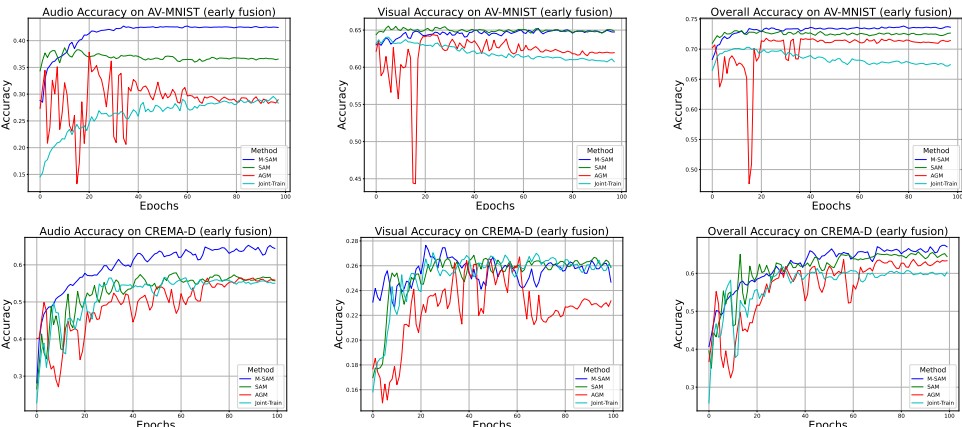

Figure 5: **Performance comparisons of early fusion settings:** Joint-Trained, AGM, OGM-GE, SAM, and our proposed Modality-Aware SAM, on the AV-MNIST and CREMA-D datasets. Viewing this in color is recommended.

values. In contrast, SAM is explicitly designed to operate on the optimization landscape by directly utilizing loss values. This fundamental difference raises a critical challenge: how can we bridge the gap between modality-specific performance contributions and the loss landscape decomposition during the training process?

To address this, we propose leveraging the Shapley value in Eq. 17. By computing Shapley values for each modality during training, we dynamically adjust the weights associated with their losses, thereby linking the concept of modality performance to loss decomposition. This dynamic adjustment ensures that modalities contributing more to performance are appropriately emphasized in the loss computation. Specifically, the weight associated with each modality's loss is derived from its normalized Shapley value. Mathematically, this is expressed as:

$$\nu_m = \frac{\Phi_m}{\sum_{i=1}^{M} \Phi_i}, \tag{20}$$

where $M$ is the total number of modalities, $\Phi_m$ is the Shapley value of the $m$-th modality, and $\nu_m$ represents the normalized weight in Eq. 4.

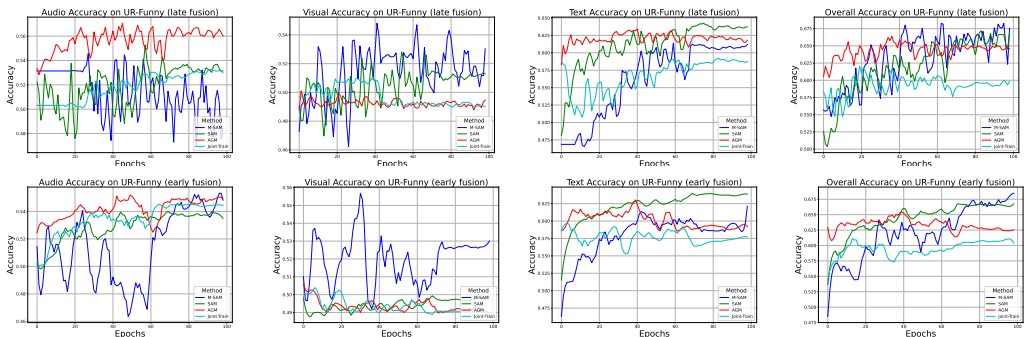

Figure 6: Performance comparisons of the Joint-Trained, AGM, SAM, and our M-SAM on UR-Funny datasets' validation set using ***late fusion*** (first row) and ***early fusion*** (second row) architecture. It is recommended to view this in color.

## C    Learning Curves

Fig. 4 and Fig. 6 present the performance comparisons of our proposed Modality-Aware SAM (M-SAM) with SAM and other state-of-the-art methods, including AGM, OMG-GE, and Joint-Train, under late fusion settings for the AV-MNIST and CREMA-D datasets and both late and early fusion scenarios of URFunny dataset, respectively. The results consistently demonstrate the superiority of M-SAM across both datasets and all metrics (audio, visual, and overall accuracy).

For the AV-MNIST dataset, M-SAM achieves the highest accuracy in all metrics. In the audio modality, M-SAM converges quickly and achieves a final accuracy of approximately 0.40, outperforming SAM (0.35), AGM, and OMG-GE, which lag significantly. In the visual modality, M-SAM demonstrates smooth and steady improvement, achieving the best accuracy of around 0.65. In contrast, SAM achieves slightly lower accuracy, and AGM and OMG-GE show notable instability. Regarding overall accuracy, M-SAM reaches approximately 0.74, clearly surpassing SAM (0.71) and significantly outperforming other methods, which fail to approach competitive levels. On the CREMA-D dataset, M-SAM maintains its superiority across all metrics. In the audio modality, M-SAM consistently achieves higher accuracy and improved stability compared to SAM, which shows oscillations during training. AGM and OMG-GE perform poorly, failing to converge effectively. In the visual modality, M-SAM achieves the highest accuracy, with SAM trailing behind and other methods struggling to maintain stability. Finally, in overall accuracy, M-SAM again emerges as the best-performing method, with the smoothest convergence and highest final accuracy.

## D    Margin of Superiority over Joint-Train baseline

To evaluate the effectiveness of each method, we report the normalized marginal improvement in accuracy over the Joint Training (JT) baseline. Specifically, we compute the percentage increase in overall accuracy ($Acc_{mm}$) for each method relative to the JT baseline using the following formulation:

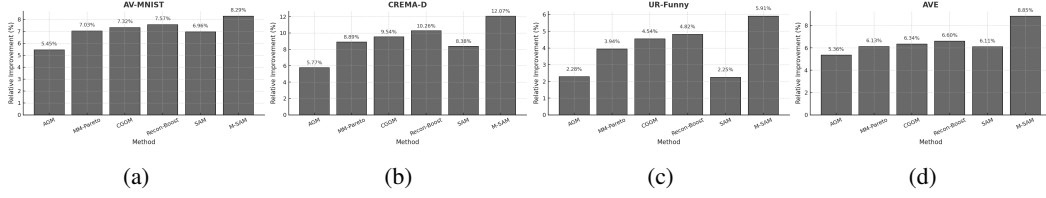

| (a) | (b) | (c) | (d) |

Figure 7: Relative improvement in multi-modal accuracy ($Acc_{mm}$) over the Joint-Training baseline on (a) AV-MNIST, (b) CREMA-D, (c) UR-Funny, and (d) AVE datasets. Our M-SAM consistently achieves the highest normalized gain, outperforming all methods.

This normalized metric allows for a fair comparison across datasets of varying base difficulty and scales, highlighting how much each method improves over standard joint optimization.

$$\Delta_{\text{rel}}(\text{Method}) = \frac{\text{Acc}_{mm}(\text{Method}) - \text{Acc}_{mm}(\text{JT})}{\text{Acc}_{mm}(\text{JT})} \times 100 \qquad (21)$$

As shown in Figure 7, our proposed *M-SAM* consistently achieves the largest relative gain across all datasets, outperforming strong baselines such as Recon-Boost and CGGM.

