# OpenReview forum: "Modality-Aware SAM: Sharpness-Aware-Minimization Driven Gradient Modulation for Harmonized Multimodal Learning"
_NeurIPS.cc/2025/Conference — NeurIPS 2025 poster_

### Official Review · Reviewer_VvWF · 2025-06-16

**Clarity:** 2
**Significance:** 1
**Originality:** 2
**Rating:** 4
**Confidence:** 3

**Summary:**

This paper focuses on the challenge of ​​modal imbalance​​ in multimodal learning.That is, existing joint training paradigms ​​couple gradient updates​​ across modality-specific encoders where the dominant modality​​ suppresses weaker modalities during training. As a result, the generalization capability of multimodal models are degraded. To this end, the authors leverage the ​​Sharpness-Aware Minimization (SAM) framework​​ to dynamically balance gradient contributions. ​​Experiments​​ on four benchmarks are carried out to validate the method’s efficacy in harmonizing modal optimization.

**Questions:**

See above.

**Ethical Concerns:**

["NO or VERY MINOR ethics concerns only"]

**Final Justification:**

The authors correctly replied my concerns. My major concerns about the soundness of theoretical analysis and reproducibility have been addressed.

**Limitations:**

yes

**Paper Formatting Concerns:**

No.

**Quality:**

2

**Strengths And Weaknesses:**

**Strength**

- The research problem is important and practical.

**Weakness**

- The paper lacks both rigious theoretical analysis or empirical evidence to demonstrate the effectiveness of M-SAM in addressing the modality imbalance problem. The motivation of M-SAM seems only supported by intuitive explaination without solid analysis. Given this point, the claim are unsubstantiated for the reviewer.

- The authors argue that the consistency between validation accuracy and training accuracy indicates better generalization. However, this is not supported by extensive evidence and seems conflict with existing works[1].

- Standard deviation or error bars are missing, which makes it difficult for the reviewer to figure out if the improvement is significant.

- The core part of the proposed method is almost identical to ImbSAM, which makes the originality of this paper not compelling enough for the reviewer.

[1] Do we need zero training loss after achieving zero training error? ICML'20

---

> ### Author Rebuttal · Authors · 2025-07-31
>
> **Response to Comment 1 (Lack of Theoretical Analysis or Empirical Evidence):**
>
> We sincerely thank the reviewer for emphasizing the importance of both empirical and theoretical validation of our method.
> To address the empirical aspect, we would like to highlight that the paper provides a broad set of experiments designed to demonstrate the effectiveness of M‑SAM in mitigating modality imbalance.
>
> For example, Figure 1 presents a qualitative comparison of the loss landscapes of M‑SAM, traditional SAM, and a strong multimodal baseline (AGM) on CREMA‑D. This figure illustrates that M‑SAM consistently converges to flatter minima, a property that is widely associated with better generalization. Complementing this, Figure 2 reports results based on the generalization‑gap metric on AV‑MNIST, where M‑SAM demonstrates a clear and consistent advantage over all benchmarks. Beyond these visualizations, Tables 1 and 2 provide an extensive quantitative comparison across four diverse and challenging datasets (AV‑MNIST, CREMA‑D, UR‑Funny, and AVE) under both early‑fusion and late‑fusion architectures for single-modal and multi-modal accuracy.
>
> These results, together with the training‑curve analyses presented in the Appendix, show that the benefits of M‑SAM hold across datasets and training regimes. Collectively, these results offer a comprehensive empirical evaluation spanning multiple settings and confirm the robustness of the proposed approach.
>
> In addition to empirical validation, the paper offers a rigorous theoretical treatment. Specifically, we derive a closed‑form mathematical representation for the M‑SAM optimization target (Eq. 12), providing a principled foundation for the method.
>
> Furthermore, our detailed analyses presented in the responses to Comments 1 and 2 of Reviewers **oFnA** and **sAY6**. Those responses address concerns about convergence proof, computation analysis of MSAM, and proof of the fact that the proposed update rule remains stable and effective even in the presence of oscillating dominant modality selection.
>
> We would like to mention that these analyses complement the empirical findings and reinforce that the contributions of M‑SAM are both theoretically sound and practically impactful.
>
>
> **Response to Comment 2 (Validation-Training Accuracy Consistency and Generalization):**
>
> We would like to thank you for your thoughtful comment. Previous studies, including \cite{Ishida}, the paper that you mentioned in your comment,  have firmly established that the gap between training and testing performance is an important indicator of the geometry of the solution found during training. For example, \cite{Keskar} demonstrates that when a training process results in a significant gap between training and testing accuracies, it signals that the network has converged to a sharp minimum, characterized by a significant number of large positive eigenvalues in $\\nabla^2 f$. Such a geometry indicates a higher sensitivity to perturbations and typically results in lower inference accuracy on unseen data.
>
> Moreover, the relation between sharp minima and reduced generalization ability is a well-established topic that has been extensively investigated, from the pioneering work of Hochreiter and Schmidhuber \cite{Hochreiter_Schmidhuber} to more recent studies on robust optimization and Sharpness-Aware Minimization \cite{SAM_paper}.
>
> We would like to highlight that in line 58, it clearly says "... huge gap shows it tends to overfit.." which is a highly compatible to what the paper you mentioned, \cite{Ishida}, elaborate on its Fig.1, when it discussed the generalization gap in phase A and phase B.
>
>
> [Ishida]: "Do we need zero training loss after achieving zero training error?" Ishida, et al. (ICML2020)
>
> [Keskar]: "On Large-Batch Training for Deep Learning: Generalization Gap and Sharp Minima. " Keskar, et al. (ICLR 2017).
>
> [Hochreiter_Schmidhuber]: "Flat minima.", Hochreiter, Sepp, and Jürgen Schmidhuber. Neural computation (1997).
>
> [SAM_paper]: "Sharpness-aware minimization for efficiently improving generalization." , Foret, et al. (ICLR 2021).
>
>
> **Response to Comment 2 (Missing standard deviations or error bars in the results):**
>
> We thank the reviewer for highlighting the importance of reporting variability.
> We would like to mention that the values reported in the tables for M‑SAM correspond to the mean accuracy over five runs with seeds 1, 2, 3, 4, and 5. Also, we would like to clarify that our decision to report results based on just mean accuracy (without SD) was made to align with the result presentation approach established in widely adopted baseline methods such as Reacon-Boost (Classifier-guided Gradient Modulation for Enhanced Multimodal Learning- ICML2024), CGGM (Classifier-guided Gradient Modulation for Enhanced Multimodal Learning- NIPS2024), MM-Pareto (MMPareto: Innocent Uni-modal Assistance for Enhanced Multi-modal Learning -ICLR 2024), AGM (Boosting Multi-modal Model Performance with Adaptive Gradient Modulation-CVPR2023).
>
> To address your concern, in the following tables we have included SD in our results (in Table.1, and Table.2) to more comprehensively demonstrate the reliability of our method.
>
>
> standard deviations for Table.1 (late fusion):
>
> | Model  | AV-MNIST Acc_a | AV-MNIST Acc_v | AV-MNIST Acc_mm | CREMA-D Acc_a | CREMA-D Acc_v | CREMA-D Acc_mm | UR-Funny Acc_a | UR-Funny Acc_v | UR-Funny Acc_t | UR-Funny Acc_mm | AVE Acc_a | AVE Acc_v | AVE Acc_mm |
> |--------|----------------|----------------|-----------------|----------------|----------------|-----------------|----------------|----------------|----------------|-----------------|-----------|-----------|-------------|
> | **M-SAM (SD)** | 0.36 | 0.44 | 0.36 | 0.66 | 0.65 | 0.46 | 0.99 | 1.01 | 1.13 | 1.00 | 0.44 | 0.26 | 0.34 |
>
>
>
> standard deviations for Table.2 (early fusion):
>
> | Model  | AV-MNIST Acc_a | AV-MNIST Acc_v | AV-MNIST Acc_mm | CREMA-D Acc_a | CREMA-D Acc_v | CREMA-D Acc_mm | UR-Funny Acc_a | UR-Funny Acc_v | UR-Funny Acc_t | UR-Funny Acc_mm | AVE Acc_a | AVE Acc_v | AVE Acc_mm |
> |--------|----------------|----------------|-----------------|----------------|----------------|-----------------|----------------|----------------|----------------|-----------------|-----------|-----------|-------------|
> | **M-SAM (SD)** | 0.49 | 0.32 | 0.31 | 0.52 | 0.53 | 0.55 | 1.12 | 0.92 | 0.89 | 1.04 | 0.40 | 0.42 | 0.49 |
>
> **Response to Comment 4 (Originality and Similarity to ImbSAM):**
>
> We would like to thank the reviewer for this thoughtful comment. As the authors transparently acknowledged in Sec. 3.5, the idea of this paper was partly inspired by (Zhou et al., 2023), i.e., the Imb-SAM framework. This, by no means, signifies that we forcibly applied Imb-SAM to our approach. The high-level differences between our approach and Imb-SAM are two-fold: (i) loss computation and decomposition (explained in Sec. 3.2), (ii) considering the dynamics of the learning process, i.e., where and how to apply the SAM-based approach (explained in Sec. 3.3). Using the intuition from Sec 3.2 and Sec. 3.3, we arrive at Eq. (12), whose final mathematical form happens to resemble that of Imb‑SAM, but the derivation, underlying objectives, and dynamics that lead to it are fundamentally different, as explained above.

---

> > ### Comment · Reviewer_VvWF · 2025-08-06
> > **Follow up**
> >
> > I appreciate the efforts the authors made to address my concerns. I spent more time in reviewing the original results and the conversation between authors and other reviewers. And now I have a clearer understanding of the theoretical analysis and its significance. This addtional context has helped me to better understand these results. I especially admire the additional standard deviations. The authors are encouraged to include them in the final version (maybe appendix). In my opinion, reporting the mean and standard deviation would be necessary especially for small and middle-sized datasets. It is suprising for me that many previous publications miss these results, but should not be encourged. l am now inclined to vote for accepting the paper. Good luck!

---

> > > ### Author Response · Authors · 2025-08-06
> > >
> > > Thank you for taking the time to revisit our paper and engage with the discussion. We are glad the clarifications and additional results helped. We also agree that reporting standard deviations is important and will include them in the final version as suggested.
> > > We truly appreciate your thoughtful reconsideration.

---

### Official Review · Reviewer_sAY6 · 2025-06-22

**Clarity:** 3
**Significance:** 3
**Originality:** 2
**Rating:** 4
**Confidence:** 4

**Summary:**

This paper addresses the important problem of modality imbalance in multi-modal learning. In current joint learning frameworks, the gradients of different modality encoders are coupled, which often leads to the dominant modality overpowering the fusion process and ultimately limiting overall performance. The authors incorporate SAM to address this issue. Comprehensive experiments demonstrate the effectiveness of the proposed method.

**Questions:**

Please see the weakness above.

**Ethical Concerns:**

["NO or VERY MINOR ethics concerns only"]

**Final Justification:**

This paper explores an important problem and conduct various experiments. However, my primary concern is the computational efficiency as mentioned in W.1. Nevertheless, I lean towards accept.

**Limitations:**

Yes, the authors have adequately addressed the limitations and potential negative societal impact of their work in the appendix section.

**Paper Formatting Concerns:**

No major formatting issues in this paper.

**Quality:**

3

**Strengths And Weaknesses:**

The strengths are listed as follows:

- The paper offers a novel perspective on mitigating modality imbalance by introducing SAM into the multi-modal learning community.

- M-SAM achieves consistent improvements over state-of-the-art methods across several benchmarks (e.g., AV-MNIST, CREMA-D, UR-Funny, AVE) in both single- and multi-modal settings. Its focus on flatter minima leads to better generalization and reduced overfitting, as shown by the normalized overfitting gap and loss landscape visualizations.

However, I have the following concerns:

- From my perspective, SAM-based methods such as ImbSAM and CCSAM face a trade-off between computational efficiency and control over the loss landscape. I am curious how the proposed M-SAM balances these two aspects.

- Although the method shows strong empirical performance, it lacks formal guarantees or deeper theoretical analysis on convergence or generalization, such as those provided by FocalSAM, especially under varied modality dominance patterns or real-world noisy scenarios.

---

> ### Author Rebuttal · Authors · 2025-07-30
>
> **Response to Comment 1 (computational efficiency):**
>
> We thank the reviewer for raising this point.
> Unlike Imb‑SAM and CCSAM (which address class imbalance), M‑SAM targets modality imbalance. Its complexity depends only on the number of input modalities $k$, not on the number of classes.
>
> In each iteration, M‑SAM computes Shapley weights using all non‑empty subsets of modalities, excluding only the trivial all‑zero subset.
> This introduces $2^k-2$ additional forward passes over SAM, with no additional backward passes or memory overhead, since these extra passes are used solely for weight estimation and do not store activations and gradients.
>
> In the paper we evaluated M-SAM using $4$ famous and widely used datasets in the field:
>
> $k=2$ (e.g., AV‑MNIST, CREMA‑D): extra forward passes = $2^2-2=2$
>
> $k=3$ (e.g., UR‑Funny): extra forward passes = $2^3-2=6$
>
> | Method          | Forward | Backward |
> | --------------- | ------- | -------- |
> | SAM             | 2       | 2        |
> | M‑SAM (\$k=2\$) | 4       | 2        |
> | M‑SAM (\$k=3\$) | 8       | 2        |
>
> M‑SAM introduces only a modest additional cost:
> for $k=2$ modalities, the forward‑to‑backward ratio changes from $(2+2)/(2+2)=1$ in SAM to $(4+2)/(2+2)=1.5$,
> and for $k=3$ modalities it becomes $(8+2)/(2+2)=2.5$,
> while the number of backward passes and memory requirements remain exactly the same as SAM.
>
> I would like to emphasize that, this overhead is entirely independent of the number of dataset classes,
> and in practice the number of modalities rarely grows beyond $2$ or $3$, so the additional cost remains small.
>
> **Response to Comment 2 (Convergence and Theoretical Guarantees):**
>
> We thank the reviewer for this valuable comment highlighting the need for a deeper theoretical analysis.
> To address this concern, we provide a convergence proof for the proposed M‑SAM method.
> Under standard smoothness and bounded gradient assumptions, we show that the average gradient norm decreases at a rate of $\mathcal{O}\big(\tfrac{\log T}{\sqrt{T}}\big)$, which matches the classical convergence behavior of SGD and SAM.
> This result formally guarantees the stability and convergence of M‑SAM even under varying modality dominance patterns.
>
>
> Under the assumptions of:
>
> 1. $\\mathcal{L}(\\theta_t) = \\sum_{m=1}^{M} \\nu_m \\mathcal{L}_m(\\theta_t)$ and its gradient is bounded $K$-smooth ($K$-Lipschitz), i.e., $\\| \\nabla \\mathcal{L}(\\theta_t) - \\nabla\\mathcal{L} (\\hat \\theta_t) \\| \\leq K \\| \\theta_t - \\hat \\theta_t\\|$,
>
> $\\forall (\\theta_t, \\hat \\theta_{t})$, that
>     $
>     \\|\\nabla \\mathcal{L}(\\theta_t)\\| \\leq G_{\\text{max}}.
>     $
>
> 2.
>     Learning rate
>     $
>     \\eta_t = \\frac{\\eta_0}{\\sqrt{t}}
>     $
>     and perturbation
>     $
>     \\rho_t = \\frac{\\rho_0}{\\sqrt{t}}.
>     $
>
> Then when $T$ is large enough the M-SAM updates satisfy:
>
> \\[
> \\frac{1}{T} \\sum_{t=1}^{T} \\|\\nabla \\mathcal{L}(\\theta_t)\\|^2
> \\leq \\mathcal{O}\\left(\\frac{\\log T}{\\sqrt{T}}\\right).
> \\]
>
> Before going through the proof, we would like to mention that, for tractability in the theoretical analysis, we assumed a smooth decay schedule \( \eta_t = \eta_0/\sqrt{t} \). In practice, our implementation uses a step-wise decay (multiplying the learning rate by $0.1$ every $70$ iterations), which decreases $ \\eta_t $ even faster. This practical schedule still satisfies the conditions required for convergence and does not weaken the theoretical guarantee.
>
> considering $d_t = \\theta_{t+1} - \\theta_t = -\\eta_t \\nabla \\mathcal{L} (\\hat{\\theta_t})$ and $\\hat{\\theta_t} = \\theta_t + \\rho_t \\frac{\\nabla_{m^\\ast} \\mathcal{L}(\\theta_t)}{\\|\\nabla_{m^\\ast} \\mathcal{L}(\\theta_t)\\|}$; where $\\nabla_{m^\\ast}$ denotes the gradient with respect to the dominant modality that generated by backpropagation of loss value corresponding to the dominant modality (Eq. 4 in the paper).
>
> Using descent Lemma for a $k$-smooth function we can write:
> \\begin{equation}
>      \\begin{aligned}
> \\mathcal{L}(\\theta_{t+1}) - \\mathcal{L}(\\theta_t) \\leq
> & -\\eta_t \\langle \\nabla \\mathcal{L}(\\theta_t), \\theta_{t+1} - \\theta_t \\rangle + \\frac{K\\eta_t^2}{2} \\|\\theta_{t+1} - \\theta_t\\|^2 = \\\\
> & -\\eta_t \\langle \\nabla \\mathcal{L}(\\theta_t), \\nabla \\mathcal{L} (\\hat \\theta_t) \\rangle + \\frac{K\\eta_t^2}{2} \\|\\nabla \\mathcal{L} (\\hat \\theta_{t})\\|^2 = \\\\
> &-\\eta_t \\langle \\nabla \\mathcal{L}(\\theta_t), \\nabla \\mathcal{L} (\\theta_{t}) -\\nabla \\mathcal{L}(\\theta_t) + \\nabla \\mathcal{L} (\\hat \\theta_t) \\rangle + \\frac{K \\eta_t^2}{2} \\|\\nabla \\mathcal{L} (\\hat \\theta_t)\\|^2 = \\\\
> & - \\eta_t \\|\\nabla \\mathcal{L}(\\theta_t)\\|^2 - \\eta_t \\langle \\nabla \\mathcal{L}(\\theta_t), \\nabla \\mathcal{L} (\\hat \\theta_{t}) - \\nabla \\mathcal{L}(\\theta_t) \\rangle  + \\frac{K \\eta_t^2}{2} (\\|\\nabla \\mathcal{L} (\\hat\\theta_{t})\\|^2) \\leq \\\\
> &-\\eta_t \\|\\nabla \\mathcal{L}(\\theta_t)\\|^2 + \\eta_t \\| \\nabla \\mathcal{L}(\\theta_t)\\| \\|\\nabla \\mathcal{L} (\\theta_{t}) - \\nabla \\mathcal{L}(\\hat\\theta_t) \\| + \\frac{K \\eta_t^2}{2}  (\\|\\nabla \\mathcal{L} (\\hat\\theta_{t})\\|^2) \\leq \\\\
> &-\\eta_t \\|\\nabla \\mathcal{L}(\\theta_t)\\|^2 + K\\eta_t \\| \\nabla \\mathcal{L}(\\theta_t)\\| \\|\hat\\theta_{t} - \\theta_t \\| + \\frac{K \\eta_t^2}{2}  (\\|\\nabla \\mathcal{L} (\\hat\\theta_{t})\\|^2) = \\\\
> &-\\eta_t \\|\\nabla \\mathcal{L}(\\theta_t)\\|^2 + K\\eta_t \\| \\nabla \\mathcal{L}(\\theta_t)\\| \\| \\rho_t \\frac{\\nabla_{m^\\ast} \\mathcal{L}(\\theta_t)}{\\|\\nabla_{m^\\ast} \\mathcal{L}(\\theta_t)\\|}\\|^2) + \\frac{K \\eta_t^2}{2}  (\\|\\nabla \\mathcal{L} (\\hat\\theta_{t})\\|^2) = \\\\
> &-\\eta_t \\|\\nabla \\mathcal{L}(\\theta_t)\\|^2 + K\\eta_t \\rho_t^2 \\| \\nabla \\mathcal{L}(\\theta_t)\\|  + \\frac{K \\eta_t^2}{2}  \\|\\nabla \\mathcal{L} (\\hat\\theta_{t})\\|^2 \\leq \\\\
> &-\\eta_t \\|\\nabla \\mathcal{L}(\\theta_t)\\|^2 + K\\eta_t \\rho_t^2 G_{max}  + \\frac{K \\eta_t^2}{2}  G_{max}^2
>      \\end{aligned}
> \\end{equation}
>
>
>
> by rearranging the inequality, it will be as:
>
> \\begin{equation}
>     \\begin{aligned}
>         &\\eta_t \\|\\nabla \\mathcal{L}(\\theta_t)\\|^2 \\leq \\mathcal{L}(\\theta_{t}) - \\mathcal{L}(\\theta_{t+1})+K\\eta_t \\rho_t^2 G_{max}  + \\frac{K \\eta_t^2}{2}  G_{max}^2
>     \\end{aligned}
> \\end{equation}
>
> considering definition of $\\eta_{t} = \\frac{\\eta_{0}}{\\sqrt{t}}$ and taking summation over all iterations on the left side of the above inequality, we can define a lower bound as follows:
>
> \begin{equation}
> \frac{\eta_{0}}{\sqrt{T}}\sum_{t=1}^{T} \\|\nabla \mathcal{L}(\theta_t)\\|^2 \leq
> \sum_{t=1}^{T} \eta_t \\|\nabla \mathcal{L}(\theta_t)\\|^2 \leq
> \sum_{t=1}^{T} \big(\mathcal{L}(\theta_{t}) - \mathcal{L}(\theta_{t+1}) \big) + K \eta_0 \rho_0^2 G_{max} \sum_{t=1}^{T} \frac{1}{t\sqrt{t}} + \frac{k \eta_0^2 G_{max}^2}{2} \sum_{t=1}^{T} \frac{1}{t}
> \end{equation}
>
> using telescope sum properties and considering $0 \\leq \\mathcal{L}(\\theta_t)  \\forall t :$
>
> \begin{equation}
>         \sum_{t=1}^{T} \big(\mathcal{L}(\theta_{t}) - \mathcal{L}(\theta_{t+1}) \big) = \mathcal{L}(\\theta_1) - \mathcal{L}(\\theta_{T+1})
>         \leq \mathcal{L}(\\theta_1)
> \end{equation}
>
>
> So we can rewrite the main inequality as follows:
>
> \begin{equation}
> \frac{1}{T}\sum_{t=1}^{T} \\|\nabla \mathcal{L}(\theta_t)\\|^2 \leq
>  \frac{\mathcal{L}(\\theta_1)}{\sqrt{T}} + \frac{K \rho_0^2 G_{max}}{\sqrt{T}} \sum_{t=1}^{T} \frac{1}{t\sqrt{t}} + \frac{K \eta_0 G_{max}^2}{2\sqrt{T}} \sum_{t=1}^{T} \frac{1}{t}
> \end{equation}
>
>
> Considering $\sum_{t=1}^{T} \frac{1}{t} \leq 1+\log T$ and the fact that $\sum_{t=1}^{T} \frac{1}{t^{3/2}}$ is a $p$‑series with $p=\tfrac{3}{2}(>1)$, its partial sum remains bounded, so the only term on the right-hand side that confine the asymptotic convergence rate is the harmonic term, which results in an overall rate of $\mathcal{O}\big(\tfrac{\log T}{\sqrt{T}}\big)$.
>
>
> \\[
> \\frac{1}{T} \\sum_{t=1}^{T} \\|\\nabla \\mathcal{L}(\\theta_t)\\|^2
> \\leq \\mathcal{O}\\left(\\frac{\\log T}{\\sqrt{T}}\\right).
> \\]
>
> Since the average gradient norm $\frac{1}{T}\sum_{t=1}^{T}\\|\nabla \mathcal{L}(\theta_t)\\|^2$ decreases at a rate of $\tfrac{\log T}{\sqrt{T}}$ and tends to $0$ as $T$ grows, the updates approach a stationary point, which by definition establishes convergence (as in standard SGD and traditional SAM).
>
> This bound, $\mathcal{O}\big(\tfrac{\log T}{\sqrt{T}}\big)$, matches the standard non‑convex convergence rate of SGD with a decaying learning rate schedule (where the logarithmic factor comes from $\sum 1/t$), so the proposed M‑SAM method has the same order of convergence as classical SGD and traditional SAM.

---

> > ### Comment · Reviewer_sAY6 · 2025-08-04
> >
> > Thanks for the author's efforts during rebuttal. The authors are encouraged to include them in the revised version.

---

> > > ### Author Response · Authors · 2025-08-04
> > >
> > > Thank you for your thoughtful review and for engaging with our rebuttal. We're glad the clarifications were helpful and will make sure to include them clearly in the revised version to improve the overall presentation and completeness of the work.

---

### Official Review · Reviewer_oFnA · 2025-06-29

**Clarity:** 3
**Significance:** 2
**Originality:** 3
**Rating:** 4
**Confidence:** 4

**Summary:**

This paper proposes Modality-Aware Sharpness-Aware Minimization (M-SAM), a novel optimizer-level approach for harmonizing multimodal learning by extending Sharpness-Aware Minimization (SAM). M-SAM focuses on guiding optimization to flatter minima by dynamically favoring the dominant modality per mini-batch using Shapley value decomposition and per-modality gradient modulation. The method is evaluated across four benchmark multimodal datasets under both early and late fusion setups and achieves superior accuracy and generalization compared to a wide range of recent state-of-the-art methods.

**Questions:**

* It would be helpful if the authors could clarify whether the proposed approach is supported by theoretical analysis or formal guarantees.
* What are the implications of dominant modality shifts across training phases or mini-batches?
* The writting issue should be fixed.

**Ethical Concerns:**

["NO or VERY MINOR ethics concerns only"]

**Final Justification:**

The paper is now well-developed and supported by solid theoretical contributions.Based on the overall quality of the work, I recommend that this paper be accepted for publication.

**Limitations:**

yes

**Paper Formatting Concerns:**

None.

**Quality:**

3

**Strengths And Weaknesses:**

**Strength:**
* By using Shapley values to identify and prioritize the dominant modality's generalization, the paper proposes a unique solution to the pervasive issue of modality imbalance and uncoordinated convergence in multimodal learning.
* M-SAM outperforms strong baselines such as CGGM, AGM, Recon-Boost, and MM-Pareto in almost all cases across all benchmark datasets.
* The proposed method performs well in both early and late fusion settings, showing its robustness and broad applicability.

**Weaknesses:**
* While intuitively and empirically strong, the paper lacks theoretical analysis or convergence guarantees under the M-SAM optimization dynamics, especially with selective gradient application.
* The paper states M-SAM "detects and prioritizes the dominant modality at each iteration and mini-batch". However, the implications of a dominant modality potentially shifting or oscillating across different training phases or mini-batches are not thoroughly discussed.
* There are still some minor issues in the writing that should be addressed for improved clarity and professionalism. For example, neither Table 1 nor Table 2 is referenced in the main text. The subsection titles in Section 4 use an unusual numbering format (e.g., 4.0.x), which is inconsistent with standard academic formatting. Figure 2 is cited before Figure 1, which disrupts the logical flow.
* Algorithm 1 is poorly typeset and hard to read. A cleaner format or pseudocode would help comprehension.

---

> ### Author Rebuttal · Authors · 2025-07-31
>
> **Response to Comment 1 (Theoretical Convergence Guarantees under the M-SAM optimization dynamics):**
>
> We would like to sincerely thank the reviewer for raising the concern about the convergence behavior of M‑SAM under its dynamic updates.
>
> Through the following proof we will show that the convergence wont be affected by the choice of the dominant modality gradient at each step. Even if there is approximation error in Shapley-based modality selection, the updates remain stable and converge.
>
> Under the assumptions of:
>
> 1. $\\mathcal{L}(\\theta_t) = \\sum_{m=1}^{M} \\nu_m \\mathcal{L}_m(\\theta_t)$ and its gradient is bounded $K$-smooth ($K$-Lipschitz), i.e., $\\| \\nabla \\mathcal{L}(\\theta_t) - \\nabla\\mathcal{L} (\\hat \\theta_t) \\| \\leq K \\| \\theta_t - \\hat \\theta_t\\|$,
>
> $\\forall (\\theta_t, \\hat \\theta_{t})$, that gradient is bounded:
>     $
>     \\|\\nabla \\mathcal{L}(\\theta_t)\\| \\leq G_{\\text{max}}.
>     $
>
> 2.
>     Learning rate
>     $
>     \\eta_t = \\frac{\\eta_0}{\\sqrt{t}}
>     $
>     and perturbation
>     $
>     \\rho_t = \\frac{\\rho_0}{\\sqrt{t}}.
>     $
>
> Then when $T$ is large enough the M-SAM updates satisfy:
>
> \\[
> \\frac{1}{T} \\sum_{t=1}^{T} \\|\\nabla \\mathcal{L}(\\theta_t)\\|^2
> \\leq \\mathcal{O}\\left(\\frac{\\log T}{\\sqrt{T}}\\right).
> \\]
>
> (Before going through the proof, we would like to mention that, for tractability in the theoretical analysis, we assumed a smooth decay schedule $\\eta_t = \\eta_0/ \\sqrt{t}$. In practice, our implementation uses a step-wise decay (multiplying the learning rate by $0.1$ every $70$ iterations), which decreases $ \\eta_t $ even faster. This practical schedule still satisfies the conditions required for convergence and does not weaken the theoretical guarantee.)
>
> Considering $d_t = \\theta_{t+1} - \\theta_t = -\\eta_t \\nabla \\mathcal{L} (\\hat{\\theta_t})$ and $\\hat{\\theta_t} = \\theta_t + \\rho_t \\frac{\\nabla_{m^\\ast} \\mathcal{L}(\\theta_t)}{\\|\\nabla_{m^\\ast} \\mathcal{L}(\\theta_t)\\|}$; where $\\nabla_{m^\\ast}$ denotes the gradient with respect to the dominant modality, generated by backpropagation of loss value corresponding to the dominant modality, $m^\\ast$ (Eq. 4 in the paper).
>
> Using descent Lemma for a $k$-smooth function we can write:
> \\begin{equation}
>      \\begin{aligned}
> \\mathcal{L}(\\theta_{t+1}) - \\mathcal{L}(\\theta_t) \\leq
> & -\\eta_t \\langle \\nabla \\mathcal{L}(\\theta_t), \\theta_{t+1} - \\theta_t \\rangle + \\frac{K\\eta_t^2}{2} \\|\\theta_{t+1} - \\theta_t\\|^2 = \\\\
> & -\\eta_t \\langle \\nabla \\mathcal{L}(\\theta_t), \\nabla \\mathcal{L} (\\hat \\theta_t) \\rangle + \\frac{K\\eta_t^2}{2} \\|\\nabla \\mathcal{L} (\\hat \\theta_{t})\\|^2 = \\\\
> &-\\eta_t \\langle \\nabla \\mathcal{L}(\\theta_t), \\nabla \\mathcal{L} (\\theta_{t}) -\\nabla \\mathcal{L}(\\theta_t) + \\nabla \\mathcal{L} (\\hat \\theta_t) \\rangle + \\frac{K \\eta_t^2}{2} \\|\\nabla \\mathcal{L} (\\hat \\theta_t)\\|^2 = \\\\
> & - \\eta_t \\|\\nabla \\mathcal{L}(\\theta_t)\\|^2 - \\eta_t \\langle \\nabla \\mathcal{L}(\\theta_t), \\nabla \\mathcal{L} (\\hat \\theta_{t}) - \\nabla \\mathcal{L}(\\theta_t) \\rangle  + \\frac{K \\eta_t^2}{2} (\\|\\nabla \\mathcal{L} (\\hat\\theta_{t})\\|^2) \\leq \\\\
> &-\\eta_t \\|\\nabla \\mathcal{L}(\\theta_t)\\|^2 + \\eta_t \\| \\nabla \\mathcal{L}(\\theta_t)\\| \\|\\nabla \\mathcal{L} (\\theta_{t}) - \\nabla \\mathcal{L}(\\hat\\theta_t) \\| + \\frac{K \\eta_t^2}{2}  (\\|\\nabla \\mathcal{L} (\\hat\\theta_{t})\\|^2) \\leq \\\\
> &-\\eta_t \\|\\nabla \\mathcal{L}(\\theta_t)\\|^2 + K\\eta_t \\| \\nabla \\mathcal{L}(\\theta_t)\\| \\|\hat\\theta_{t} - \\theta_t \\| + \\frac{K \\eta_t^2}{2}  (\\|\\nabla \\mathcal{L} (\\hat\\theta_{t})\\|^2) = \\\\
> &-\\eta_t \\|\\nabla \\mathcal{L}(\\theta_t)\\|^2 + K\\eta_t \\| \\nabla \\mathcal{L}(\\theta_t)\\| \\| \\rho_t \\frac{\\nabla_{m^\\ast} \\mathcal{L}(\\theta_t)}{\\|\\nabla_{m^\\ast} \\mathcal{L}(\\theta_t)\\|}\\|^2) + \\frac{K \\eta_t^2}{2}  (\\|\\nabla \\mathcal{L} (\\hat\\theta_{t})\\|^2) = \\\\
> &-\\eta_t \\|\\nabla \\mathcal{L}(\\theta_t)\\|^2 + K\\eta_t \\rho_t^2 \\| \\nabla \\mathcal{L}(\\theta_t)\\|  + \\frac{K \\eta_t^2}{2}  \\|\\nabla \\mathcal{L} (\\hat\\theta_{t})\\|^2 \\leq \\\\
> &-\\eta_t \\|\\nabla \\mathcal{L}(\\theta_t)\\|^2 + K\\eta_t \\rho_t^2 G_{max}  + \\frac{K \\eta_t^2}{2}  G_{max}^2
>      \\end{aligned}
> \\end{equation}
>
> _Here we would like to highlight that the step of finding the upper bound on the right-hand side of the inequality is the only place where the dynamic nature of modality selection could potentially affect convergence. However, it is clearly shown that the convergence is independent of_ $\\nabla_{m^\\ast} \\mathcal{L}(\\theta_t)$. *This is because the upper bound,* $\\| \\rho_t \\frac{\\nabla_{m^\\ast} \\mathcal{L}(\\theta_t)}{\\|\\nabla_{m^\\ast} \\mathcal{L}(\\theta_t)\\|}\\|^2$, *depends only on* $\\rho_t$_, *and not on the direction of the selected modality’s gradient.* Hence, the convergence is independent of modality selection at each iteration.
>
> By rearranging the inequality, it will be as:
>
> \\begin{equation}
>     \\begin{aligned}
>         &\\eta_t \\|\\nabla \\mathcal{L}(\\theta_t)\\|^2 \\leq \\mathcal{L}(\\theta_{t}) - \\mathcal{L}(\\theta_{t+1})+K\\eta_t \\rho_t^2 G_{max}  + \\frac{K \\eta_t^2}{2}  G_{max}^2
>     \\end{aligned}
> \\end{equation}
>
> considering definition of $\\eta_{t} = \\frac{\\eta_{0}}{\\sqrt{t}}$ and taking summation over all iterations on the left side of the above inequality, we can define a lower bound as follows:
>
> \begin{equation}
> \frac{\eta_{0}}{\sqrt{T}}\sum_{t=1}^{T} \\|\nabla \mathcal{L}(\theta_t)\\|^2 \leq
> \sum_{t=1}^{T} \eta_t \\|\nabla \mathcal{L}(\theta_t)\\|^2 \leq
> \sum_{t=1}^{T} \big(\mathcal{L}(\theta_{t}) - \mathcal{L}(\theta_{t+1}) \big) + K \eta_0 \rho_0^2 G_{max} \sum_{t=1}^{T} \frac{1}{t\sqrt{t}} + \frac{k \eta_0^2 G_{max}^2}{2} \sum_{t=1}^{T} \frac{1}{t}
> \end{equation}
>
> using telescope sum properties and considering $0 \\leq \\mathcal{L}(\\theta_t)  \\forall t :$
>
> \begin{equation}
>         \sum_{t=1}^{T} \big(\mathcal{L}(\theta_{t}) - \mathcal{L}(\theta_{t+1}) \big) = \mathcal{L}(\\theta_1) - \mathcal{L}(\\theta_{T+1})
>         \leq \mathcal{L}(\\theta_1)
> \end{equation}
>
>
> So we can rewrite the main inequality as follows:
>
> \begin{equation}
> \frac{1}{T}\sum_{t=1}^{T} \\|\nabla \mathcal{L}(\theta_t)\\|^2 \leq
>  \frac{\mathcal{L}(\\theta_1)}{\sqrt{T}} + \frac{K \rho_0^2 G_{max}}{\sqrt{T}} \sum_{t=1}^{T} \frac{1}{t\sqrt{t}} + \frac{K \eta_0 G_{max}^2}{2\sqrt{T}} \sum_{t=1}^{T} \frac{1}{t}
> \end{equation}
>
>
> Considering $\sum_{t=1}^{T} \frac{1}{t} \leq 1+\log T$ and the fact that $\sum_{t=1}^{T} \frac{1}{t^{3/2}}$ is a $p$‑series with $p=\tfrac{3}{2}(>1)$, its partial sum remains bounded, so the only term on the right-hand side that confine the asymptotic convergence rate is the harmonic term, which results in an overall rate of $\mathcal{O}\big(\tfrac{\log T}{\sqrt{T}}\big)$.
>
>
> \\[
> \\frac{1}{T} \\sum_{t=1}^{T} \\|\\nabla \\mathcal{L}(\\theta_t)\\|^2
> \\leq \\mathcal{O}\\left(\\frac{\\log T}{\\sqrt{T}}\\right).
> \\]
>
> Since the average gradient norm $\frac{1}{T}\sum_{t=1}^{T}\\|\nabla \mathcal{L}(\theta_t)\\|^2$ decreases at a rate of $\tfrac{\log T}{\sqrt{T}}$ and tends to $0$ as $T$ grows, the updates approach a stationary point, which by definition establishes convergence (as in standard SGD and traditional SAM).
>
> This bound, $\mathcal{O}\big(\tfrac{\log T}{\sqrt{T}}\big)$, matches the standard non‑convex convergence rate of SGD with a decaying learning rate schedule (where the logarithmic factor comes from $\sum 1/t$). Thus, the proposed M-SAM method preserves the same theoretical convergence guarantees as classical SGD and traditional SAM while offering dynamic modality-based perturbations.
>
> **Response to Comment 2 (Implications of Dominant Modality Shifts across Training Phases and Mini‑Batches):**
>
> We thank the reviewer for raising this thoughtful point about shifts in the dominant modality during training and their effect on the learning dynamics of M‑SAM.
>
> M‑SAM is designed to benefit from these shifts rather than suppress them. By re‑evaluating the dominant modality at every iteration using Shapley‑based contributions, the method dynamically allocates attention to the modality that is most informative at that stage of training.
>
> As discussed in our response to Comment 1, such alternations do not compromise stability or convergence. Instead, they guide the currently dominant modality toward flatter minima, giving other modalities freedom to contribute. When dominance oscillates between modalities, this competition prevents any single modality from monopolizing learning, leading to a more balanced and robust multimodal representation.
>
> **Response to Comment 3 (Writing Issues):**
>
> We appreciate the reviewer’s detailed feedback on clarity and presentation. We acknowledge these issues and will address them in the camera‑ready version by correcting table references, subsection numbering, figure order, and improving the formatting of Algorithm 1 for better readability.

---

> > ### Comment · Reviewer_oFnA · 2025-08-02
> > **Comment**
> >
> > I appreciate the authors' efforts in addressing my concerns. The paper is now well-developed and supported by solid theoretical contributions. Therefore, I am willing to raise my score.

---

> > > ### Author Response · Authors · 2025-08-02
> > >
> > > We appreciate your thoughtful follow up and are glad that the revisions addressed your concerns. Your feedback helped us clarify the theoretical contributions and improve the paper overall. Thank you for taking the time to review our work so carefully.

---

### Decision · Program_Chairs · 2025-09-17

**Decision:**

Accept (poster)

**Comment:**

This paper tackles the challenge of modality imbalance in multimodal learning by proposing M-SAM, a modality-aware extension of Sharpness-Aware Minimization. M-SAM identifies the dominant modality in each iteration using Shapley-based contribution analysis and modulates the gradient updates accordingly, guiding optimization toward flatter minima while mitigating the dominance effect.
The method demonstrates consistent improvements over strong baselines across multiple datasets and both early- and late-fusion settings. In the rebuttal, the authors further strengthened the work with convergence analysis and standard deviation reporting, effectively addressing concerns about efficiency, stability, and reproducibility. Overall, the paper makes a solid contribution in method design, theoretical support, and empirical validation, and I recommend acceptance.